# How the Soluble Human Leukocyte Antigen-G levels in Amniotic Fluid and Maternal Serum Correlate with the Feto-Placental Growth in Uncomplicated Pregnancies

**DOI:** 10.3390/bioengineering11050509

**Published:** 2024-05-18

**Authors:** Márió Vincze, János Sikovanyecz, Imre Földesi, Andrea Surányi, Szabolcs Várbíró, Gábor Németh, Zoltan Kozinszky, János Sikovanyecz

**Affiliations:** 1Department of Obstetrics and Gynecology, Albert Szent-Gyorgyi Medical School, University of Szeged, H-6725 Szeged, Hungary; vincze.mario92@gmail.com (M.V.); janossikovanyecz@gmail.com (J.S.J.); gaspar-suranyi.andrea@med.u-szeged.hu (A.S.); varbiro.szabolcs@med.u-szeged.hu (S.V.); nemeth.gabor@med.u-szeged.hu (G.N.); drsikovanyecz@gmail.com (J.S.); 2Department of Laboratory Medicine, Albert Szent-Gyorgyi Medical School, University of Szeged, H-6720 Szeged, Hungary; foldesi.imre@med.u-szeged.hu; 3Capio Specialized Center for Gynecology, Postgången 53, 171 45 Solna, Sweden

**Keywords:** soluble human leukocyte antigen-G, serum, amniotic fluid, fetus, placenta, sonography

## Abstract

**Introduction:** Trophoblast-derived angiogenic factors are considered to play an important role in the pathophysiology of various complications of pregnancy. Human Leukocyte Antigen-G (HLA-G) belongs to the non-classical human major histocompatibility complex (MHC-I) molecule and has membrane-bound and soluble forms. HLA-G is primarily expressed by extravillous cytotrophoblasts located in the placenta between the maternal and fetal compartments and plays a pivotal role in providing immune tolerance. The aim of this study was to establish a relationship between concentrations of soluble HLA-G (sHLA-G) in maternal serum and amniotic fluid at 16–22 weeks of gestation and the sonographic measurements of fetal and placental growth. **Materials and methods:** sHLA-G in serum and amniotic fluid, as well as fetal biometric data and placental volume and perfusion indices, were determined in 41 singleton pregnancies with no complications. The level of sHLA-G (U/mL) was tested with a sandwich enzyme-linked immunosorbent assay (ELISA) kit. **Results:** The sHLA-G levels were unchanged both in amniotic fluid and serum during mid-pregnancy. The sHLA-G level in serum correlated positively with amniotic sHLA-G level (β = 0.63, *p* < 0.01). Serum sHLA-G level was significantly correlated with abdominal measurements (β = 0.41, *p* < 0.05) and estimated fetal weight (β = 0.41, *p* < 0.05). Conversely, amniotic sHLA-G level and placental perfusion (VI: β = −0.34, *p* < 0.01 and VFI: β = −0.44, *p* < 0.01, respectively) were negatively correlated. A low amniotic sHLA-G level was significantly associated with nuchal translucency (r = −0.102, *p* < 0.05). **Conclusions:** sHLA-G assayed in amniotic fluid might be a potential indicator of placental function, whereas the sHLA-G level in serum can be a prognostic factor for feto-placental insufficiency.

## 1. Introduction

Human Leukocyte Antigen-G (HLA-G) is a member of non-classical human major histocompatibility complex class Ib molecules. HLA-G is a key molecule in maternal–fetal immune tolerance maintaining immune homeostasis [1]. It was initially described in extravillous trophoblasts (EVT), and displays different characteristics from the classical HLA molecules. The membrane-bound subtypes are HLA-G1-4, which are expressed by EVT cells in the placenta. The soluble forms (HLA-G1, -G5, -G6, and -G7) are released from the cells due to the lack of transmembrane and intracellular domains in the membrane-bound form. The soluble subtypes are produced by various cells, such as extravillous and villous cytotrophoblasts, chorion and amnion membranes, syncytiotrophoblasts, decidual stromal cells, fetal endothelial cells, and immune cells, in the placenta [2,3].

Basically, HLA-G exhibits immune tolerance through the direct inhibition of different immune-competent cells. HLA-G receptors are inhibitory receptors, such as killer cell immunoglobulin-like receptor 2DL4 (KIR2DL4) and immunoglobulin-like transcription receptor type 2 and type 4 (ILT2 and ILT4, respectively) [4]. Killer-cell immunoglobulin-like receptor (KIR) is mainly expressed on the plasma membrane of two subtypes of natural killer (NK) cells: decidual natural killer (dNK) and peripheral blood natural killer (pbNK) cells. The ligands for several KIR proteins are subsets of HLA class I molecules; therefore, KIR proteins are thought to play a significant role in the downregulation of maternal immune response. HLA-G, particularly sHLA-G isoforms, are sole ligands of KIR2DL4 [4]. Notably, HLA-G inhibits NK cell cytotoxicity and is required to protect trophoblasts against NK cell-induced lysis [5], yet the mechanism by which EVTs uniquely express HLA-G remains unknown. Since NK cells constitute the predominant lymphocyte subset in the placenta, the pro-inflammatory/pro-angiogenic outcome of the interaction between KIR2DL4 and soluble HLA-G supports the role of KIR2DL4 in the extensive remodeling of maternal vasculature during the early weeks of pregnancy [5]. The ligand–receptor interaction activates angiogenesis and vasculogenesis-related processes and inhibits the cytotoxic activity of NK cells. sHLA-G isoforms induce trophoblast invasion and the remodeling of spiral arteries by producing pro-inflammatory cytokines (IL-6, TNF-alfa) and proangiogenic factors (VEGF-A, MMP) via NK cells [1,6]. sHLA-G promotes the secretion of IL-8 in NK cells and macrophages, which are manifest in trophoblast invasion. The activation processes encompass the production of interferon- γ (IFN- γ) by non-activated NK cells, enhancing the paracrine secretion of HLA-G. The HLA-G-induced NK-derived IFN-γ is essential for decidual disorganization, vessel modification through the promotion of cell adhesion, and proliferation of smooth muscle [7].

Moreover, HLA-G interacts with ILT2 and ILT4 receptors, thereby inhibiting cytotoxicity and enhancing the production of anti-inflammatory cytokines (IL-4, IL-10, TGF-beta). ILT2 can be found on the surface of T cells, B cells, natural killer (NK) cells, and antigen-presenting cells (APCs), whereas ILT4 is exclusive to dendritic cells, macrophages/monocytes, and APCs. The HLA-G molecule binds to both ILT2 and ILT4 with high affinity, thus inhibiting maternal immune response [8]. The direct interaction of HLA-G with different immune cell subpopulations and endothelial cells is accompanied by the generation and maintenance of tolerance at different stages of immune response, e.g., differentiation, proliferation, cytolysis, and cytokine secretion. On the other hand, HLA-G can be bound to ILT2 on dNK-cells and activate the PI3K-ACT signal transmission path. This interaction stimulates the production of growth-promoting factors, such as pleiotrophin, osteoglycin, and osteopontin, which are crucial for embryonic and fetal development [1,6]. Soluble forms have an antiproliferative effect on CD8+ T lymphocytes by promoting apoptosis of these cells. Furthermore, these forms trigger the shift in Th1-Th2 and Th17-regulatory T cells to reduce a maternal immune reaction [1,2].

Some studies have revealed a reduced expression and polymorphism of HLA-G in pathological conditions during pregnancy, including preeclampsia [9] or recurrent spontaneous abortion [10], compared to healthy placentas.

Soluble HLA-G can be detected both in maternal blood [9,11,12,13,14] and amniotic fluid [15,16]. In this study, we investigated sHLA-G (sHLA-G1 and sHLA-G5) concentration in amniotic fluid and maternal serum mid-pregnancy. Considering uncomplicated pregnancies, we investigated the correlations of sHLA-G levels with fetoplacental growth during mid-pregnancy.

## 2. Materials and Methods

### 2.1. Study Design

A prospective, cross-sectional cohort study was carried out in pregnant women with amniocentesis at the Department of Obstetrics and Gynecology, University of Szeged, Hungary, between January 2021 and May 2022. The demography of our study group was homogenic (all of them were Caucasian female subjects). During the study period, all women with singleton pregnancies with an increased risk of chromosomal abnormality, in which amniocentesis (AC) was performed between 16 + 0 and 22 + 0 weeks of gestation, were recruited in our study. The indications for AC were increased nuchal translucency (NT) at first trimester scan (≥2 MoM for gestational age (GA)) (n = 2), chromosome aberration or gene disorder concerning a previous pregnancy (n = 17), and advanced maternal age (≥35 years) (n = 22).

The exclusion criteria of the study were identified as follows: multiple pregnancies, fetal or neonatal structural or genetic anomalies, improper localization of the placenta for sonography (placenta praevia, posterior placenta), pathological placentation (placenta accreta spectrum), self-reported drug, alcohol, caffeine or nicotine abuse or exposure to circulatory medication (oxerutins, calcium dobesilate), and systemic disease (e.g., essential hypertension, any type of pregestational diabetes mellitus, autoimmune disease, vasculitis, hemophylia, thrombophylia, or chronic infection).

In addition, women with complications during late pregnancy (e.g., gestational diabetes mellitus, premature rupture of membranes, preterm birth, any infection during delivery, hypertension-related diseases, placental abruption, small for gestational age at delivery, large for gestational age at delivery) were excluded from the study. Therefore, 41 pregnancies with no complication were registered in our study. Clinical and anamnestic data were collected from the medical records of subjects.

The study protocol was approved by the Clinical Research Ethics Committee of University of Szeged (date of approval: 10 February 2017; reference number: SZTE 09/2017). The study was carried out according to the principles of the Declaration of Helsinki. We obtained written informed consent from all participants.

### 2.2. Conventional Two-Dimensional (2-D) Sonographic Examinations

All pregnancies were dated by using the measurement of crown rump length (CRL) at nuchal screening. NT and anatomic assessments between 11 + 0 and 13 + 6 weeks were performed utilizing conventional methods. An ultrasound examination took place before measuring AC to determine the number of fetuses, fetal biometry, fetal anomalies, placental location, and the amount of amniotic fluid. Fetal weight was estimated according to the method of Hadlock et al. [17] after measuring the necessary sonographic parameters (biparietal diameter, head circumference, abdominal circumference, and femur length). Estimated fetal weight percentile was calculated according to local standards [18]. The ultrasound investigations were conducted by J.S. and A.S.

### 2.3. Volume Acquisition

The images used for the determination of placental volume and Three-Dimensional Power Doppler (3-DPD) indices were acquired at the time of visit. All 3D scans were performed by an A.S. Voluson 730 Expert ultrasound machine (GE Medical Systems, Kretztechnik GmbH & Co OHG, Tiefenbach, Austria) equipped with a multifrequency probe (2–5 MHz). Each sample was examined using 3D rendering mode, in which the color and gray value information was processed and combined to provide a 3D image (mode cent; smooth: 4/5; FRQ: low; quality: 16; density: 6; enhance: 16; balance: 150; filter: 2; actual power: 2 dB; pulse repetition frequency: 0.9). We used fast low-resolution acquisition to avoid any kind of artifact. The 3D static volume box was placed over the highest-villous-vascular-density zone at the insertion of the umbilical cord [19]. Each image was recovered from the disc in succession for processing. We recorded one sample from each patient during gestation.

### 2.4. Determination of Power Doppler Indices

The stored volumes were further analyzed using the virtual organ computer-aided analysis (VOCAL) program pertaining to computer software 4DView (GE Medical Systems, Zipf, Austria, version 10.4) by the same expert in 3D analysis (A.S.). The image used for recovery from the hard disc was captured and processed using a multiplanar system. The spherical sample volume was consistently 28 mL. The VOCAL program calculated the grey- and color-scale values automatically from the acquired volume of the spherical sample in a histogram in all cases. The combined use of a power Doppler with three-dimensional ultrasound provides the possibility of quantifying blood in motion within a volume of interest. Three indices were calculated, namely the vascularization index (VI), flow index (FI), and vascularization flow index (VFI), as estimates of the percentage of volume filled with detectably moving blood. VI (expressed as a percentage) is the proportion of color voxels in the studied volume, representing the proportion of blood vessels within the tissue. FI (expressed on a scale of 0–100) is the average value of all color voxels, representing the average power Doppler amplitude within blood vessels. VFI (expressed on a scale of 0–100) is the average color value of all grey and color voxels and the product of the number of color voxels, expressed as a percentage, and the relative amplitude of these voxels [20].

The intra-observer errors were evaluated by repeated measurements of 3-DPD indices at the start of the study. The intra-class correlation coefficients for all Doppler indices were excellent (0.99) in the case of all indices.

### 2.5. Procedure of Amniocentesis

The subjects were informed about the procedure and possible complications before the consent form was signed prior to the procedure. All procedures were performed by the same operating expert (J.S.) at the outpatient unit, who followed the standard protocol. Local antiseptics were applied to the skin. A 22-gauge spinal needle was inserted under continuous ultrasound guidance, and the insertion of a needle in the placenta was avoided. Amniotic fluid (8–10 mL) was taken, and the first 2 mL of each sample was discarded to prevent contamination with maternal cells. Blood-contaminated amniotic fluid was not utilized. Fetal heart rate was evaluated after the procedure, and no stillbirths or premature ruptures were observed. Following amniocentesis, anti-D immunoglobulin was administered, if necessary.

### 2.6. Samples

Amniotic fluid and maternal venous blood were collected from each subject at the time of amniocentesis. Blood samples were centrifuged at 3400 rpm for 15 min. Serum and amniotic fluid samples were stored at −80 °C until assay.

### 2.7. Enzyme-Linked Immunosorbent Assay (ELISA)

Human sHLA-G in the maternal serum and amniotic fluid were determined by ELISA. Laboratory staff members who performed the assays were blinded to pregnancy outcome, and the clinician recruiting women did not participate in analyzing the samples.

The concentration of sHLA-G was measured using Elabscience Biotechnology Corporation kits (Houston, TX, USA). The sensitivity of the assay was 0.38 ng/mL. The intra- and inter-assay coefficients of variation were <10% according to the manufacturer.

### 2.8. Data and Statistical Analysis

Statistical analyses were performed using SPSS version 23 (IBM Corp., Armonk, NY, USA). Continuous variables were expressed as mean ± standard deviation (SD) and categorical variables were expressed as numbers and percentages. The relationship between the level of sHLA-G and the other continuous variables was assessed using univariate and multivariate regression analyses characterized by the correlation coefficient (ß) and 95% confidence interval (CI). Multiple linear regression was adjusted for well-known confounders, such as maternal age, body mass index (BMI) at the time of amniocentesis, number of previous pregnancies, and gestational age at the time of amniocentesis, as these factors determine the actual placental volume and fetal weight. A paired samples t-test was applied to analyze the differences between the serum and amniotic levels of the analytes. Two-tailed statistical significance level was set at 5% and *p*-values were adjusted using Holm–Bonferroni correction for multiple comparisons.

## 3. Results

Descriptive statistics are shown in Table 1. As expected from the guidelines, amniocentesis has been proposed for pregnant women with advanced maternal age (above 35 years), previous congenital anomalies, and suspicion of aneuploidy in the current pregnancy due to high NT. In our case group, mean maternal age (33.63 years) was slightly higher compared to the national average reference age at delivery, which was 29.1 years for primiparous women and 30.4 years for the total number of pregnant women in 2021 [21]. Women who participated in our study had a mean BMI of 26.4 kg/m^2^ and were slightly overweight, and 29.3% of the participants were primiparous. The birth weight in our sample was also near to the national average of pregnant women with no complications. As expected, the birth weight of offsprings was without extremities. Ultrasound characteristics are demonstrated in Table 2. A large NT was scarcely observed in our group (n = 2). The mean gestational age at the time of amniocentesis was 18.4 weeks and the mean gestational age at the time of delivery was 39.0 weeks (range: 37.0–41.0 weeks). Typically, all the fetal sonographic measurements had no outliers and a low standard deviation (<1) explicating that data are clustered tightly around the mean.

Table 3 provides an overview of the levels of angiogenic factors. The mean concentration was 53.39 ng/mL for sHLA-G in amniotic fluid and 51.05 ng/mL for sHLA-G in serum. 

Figure 1 presents the angiogenic factor levels in body fluids, adjusted to the gestational age. sHLA-G levels were steady during the investigated gestational period. The sHLA-G level in serum was significantly associated with the sHLA-G level in amniotic fluid, in both the unadjusted (β = 0.63, 95% CI = 0.30–0.72, *p* < 0.01) and adjusted models (β = 0.66, 95% CI = 0.32–0.74, *p* < 0.01), when controlled for age, BMI, previous parity, and gestational age at amniocentesis.

Table 4 displays the sonographic correlates of sHLA-G levels. The sHLA-G serum concentration demonstrated a significant positive correlation with abdominal circumference percentiles in the unadjusted analysis (β = 0.41, 95% CI= −0.08–0.75, *p* < 0.05). Similarly, we found a significant interaction between sHLA-G levels in serum and the estimated fetal weight percentile (β = 0.41, 95% CI = −0.02–0.84, *p* < 0.05). An inverse correlation of sHLA-G level in serum with FI could be detected in the unadjusted linear regression model (β = −0.34, 95% CI = −3.58–0.46, *p* < 0.05). Reduced nuchal translucency appears to be correlated with a high sHLA-G level in amniotic fluid (β = −0.30, 95% CI = −21,73–0.04, *p* < 0.05). An indirect correlation of sHLA-G concentrations in amniotic fluid with placental perfusion could be observed [VI: univariate analysis: β = −0.34, 95% CI = −2.13–0.06, *p* < 0.05; multivariate analysis: β = −0.38, 95% CI = −2.47–0.03, *p* < 0.05. VFI: univariate analysis: β = −0.44, 95% CI = −3.28–0.63, *p* < 0.05; multivariate analysis: β = −0.51, 95% CI = −3.79–0.72, *p* < 0.05].

## 4. Discussion

HLA-G is a molecule that was first known to protection the fetus against destruction by the maternal immune system, thus critically contributing to maternal–fetal immune tolerance. Appropriate vasculogenesis and angiogenesis play an important role in placental development. The trophoblast invasion and remodeling of spiral arteries are regulated by the balance and interaction between pro-angiogenic and anti-angiogenic factors. An imbalance in angiogenic factors can lead to pathological gestational conditions, such as gestational hypertension, preeclampsia, and prematurity [14,15,22,23].

We found a positive correlation between sHLA-G levels in serum and amniotic fluid, even when adjusted for confounders. sHLA-G in maternal circulation is secreted by EVT cells, while sHLA-G expressed by the amniotic membrane is detectable in amniotic fluid, ensuring that it is involved in both local and systemic maternal immune modulation, which is in line with previous reports [15]. The association can be explained by the fact that the local and systemic immune responses are in concordance to this extent. However, why the amniotic membrane cells secrete sHLA-G into the amniotic fluid (which is in line with the sHLA-G level in serum) is not understood, but may be related to the size of the fetus. A larger area of amniotic epithelial cells may present a higher cell mass, interacting with a more immune cells. In one study, it was reported that the sHLA-G levels measured in amniotic fluid did not change during mid-trimester (14–18 weeks), and there was no difference between levels at mid-trimester and at term [15]. Nonetheless, our results on sHLA-G levels in amniotic fluid (mean: 53.39 ng/mL) differ from the findings of Kusanovic et al. [15] (median: 27.3 ng/mL), but the authors collected samples from both complicated and uncomplicated pregnancies.

Furthermore, HLA-G is primarily a checkpoint molecule. In our research, we realized that sHLA-G levels in the maternal serum and amniotic fluid are constant during mid-pregnancy, although the level in maternal serum was decreased slightly and non-significantly. Two studies [11,13] showed similar circulating maternal sHLA-G levels (43.6 ng/mL and 39.37 ng/mL, respectively) in the second trimester in uncomplicated pregnancies compared to the results in our sample (mean 51.1 ng/mL). Moreover, the maternal serum level exhibits a positive correlation with fetal growth in terms of abdominal circumference and estimated fetal weight if expressed as a percentile, but not as a crude value. This can be explained by the fact that the HLA-G levels in serum show a wide range of variety in a relatively short time period during gestation, as published by other authors [24]. Furthermore, the fetal growth percentiles in our sample were constant and independent of gestation between the 16th and 22nd weeks. However, our study supports the idea that sHLA-G in serum is able to predict the weight percentile in fetuses with normal weight during the second trimester. Moreover, a previous study has revealed that sHLA-G is significantly decreased in the maternal blood of patients with the diagnosis of fetal growth restriction [12]. The possible molecular explanation is that sHLA-G stimulates decidual NK cells, which promotes fetal growth and controls trophoblast invasion during mid-pregnancy [1,25]. According to recent studies, another explanation is that HLA-G was found to facilitate fetal growth by stimulating the secretion of growth-promoting factors (GPFs) in NK cells [25,26,27]. Moreover, based on a previous study [12], sHLA-G levels in serum peak in the second trimester, but then decline until term in both complicated and uncomplicated pregnancies.

One can speculate that a higher EVT mass can coordinate a greater number of spiral arteries, which can supply the higher demands for gas and nutrition exchange of a larger fetus. 

In contrast to our former article, where the dataset on normal and pathological pregnancies is presented [22], our current results show no association between sHLA-G levels in serum and placental volume. This can be explained by the fact that our new sample comprised only asymptomatic, uncomplicated pregnancies with no extreme values, which should correspond to pathological pregnancies.

HLA-G is unusual among HLA molecules in its unique pattern of expression in healthy individuals [28]. Similar to our former results for healthy and complicated pregnancies [22], the sHLA-G level in amniotic fluid exhibited a negative correlation with the vascular perfusion of placenta. In this study, we explored the inverse correlation of HLA-G level in maternal serum and the placental blood perfusion with regard to healthy pregnancies. We assume that sHLA-G secretion may be reactive to the reduced blood flow and vascular network caused by the expanding placenta, since sHLA-G facilitates the process of vasculogenesis [24]. The expansion of placental volume may outweigh the increase in capillary branching. In order to evaluate the placental perfusion, we used “Mercé-type sonobiopsy”, which is a reproducible and validated method [29,30], and by obtaining a representative sample of the placental tree, this can be applied throughout the entire pregnancy, unlike other methods [20], when the entire placenta needs to be visualized [31,32]. It can be also hypothesized that the HLA-G-specific interactions of decidual natural killer cells and macrophages lead to the secretion of angiogenic and pro-inflammatory factors implicated in vascular remodeling and determine the extent of trophoblast invasion, which is connected to lower perfusion indices, as previously detected by ultrasound examinations.

Interestingly, we unraveled an indirect correlation between NT measurements during 11–14 weeks of gestation and sHLA-G levels in amniotic fluid during 16 and 22 weeks of gestation. Our results support the assumption that low vascularization and perfusion of the placenta with high sHLA-G levels in amniotic fluid during the second trimester is preceded by an impaired placental 3D-PD indices and higher NT in the first trimester. This is in line with the results of Metzenbauer et al. [33], who found that trisomies are associated with a higher NT and smaller placental volume between 11 and 14 weeks of gestation. Based on the literature, we can confirm that a low sHLA-G expression is commonly associated with implantation failure and miscarriage [10,34], as was also confirmed by our results, since an obvious correlation exists between increased nuchal translucency and fetal loss [35]. One can speculate that a lower sHLA-G level measured in the amniotic fluid may play a role in embryonic/fetal development.

Our study is focused on amniotic samples deriving from a single center, where the ultrasound measurements were performed by two highly skilled sonographers.

In conclusion, sHLA-G is unchanged in the serum and amniotic fluid during the second trimester in high-risk pregnancies with no complications. Notably, our results indicate an association between the decreased amniotic concentration of sHLA-G and nuchal translucency. Furthermore, serum sHLA-G is linked to fetal growth within the normal range, which might also be explained by embryonic or fetal developmental processes. However, both results require further studies with larger sample sizes. Proper calculations of sample size may validate these findings. It is of clinical importance that sHLA-G in amniotic fluid and maternal serum can be utilized as a potent biomarker of placental perfusion. To set up sHLA-G as a meaningful clinical biomarker, it is of paramount importance that we trace the structural diversity back to its tolerance-mediating functions.

## Figures and Tables

**Figure 1 bioengineering-11-00509-f001:**
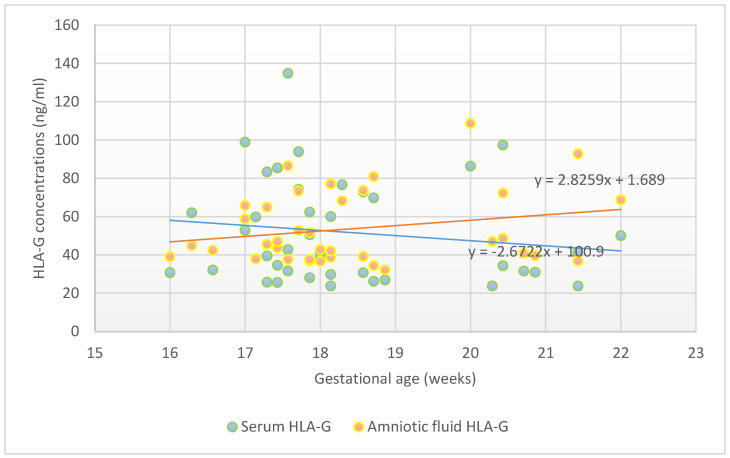
HLA-G concentrations between 16th and 22nd gestational weeks.

**Table 1 bioengineering-11-00509-t001:** Clinical and obstetric data of pregnant women with amniocentesis (N = 41).

Maternal Age (Years) *	33.63 ± 6.51
Number of nulliparous women in the study **	12 (29.3)
BMI at the time of genetic consultation (kg/m^2^) *	26.35 ± 6.19
Birth weight (grams) *	3351.22 ± 370.10
Birth weight (percentile) *	54.34 ± 24.39
Gestational age at the time of delivery (weeks) *	39.01 ± 1.32

* Continuous variables are displayed as mean ± standard deviation (SD). ** Categorical variables are presented as number and %.

**Table 2 bioengineering-11-00509-t002:** Sonographic data (N = 41).

Data on ‘Genetic’ Ultrasound Examination in the First Trimester	
NT (mm)	1.88 ± 0.66
CRL at NT (mm)	63.90 ± 6.54
Gestational age at nuchal translucency (weeks)	12.62 ± 0.55
Fetal biometry at the time of amniocentesis	
Gestational age at the time of amniocentesis (weeks)	18.37 ± 1.49
Head circumference (mm)	153.62 ± 15.89
Head circumference (percentile)	56.70 ± 29.21
Abdominal circumference (mm)	134.10 ± 17.06
Abdominal circumference (percentile)	55.25 ± 27.57
Femur length (mm)	27.71 ± 4.99
Femur length (percentile)	57.54 ± 27.46
Estimated fetal weight (grams)	260.71 ± 81.26
Estimated fetal weight (percentile)	53.50 ± 26.11
Placental sonography	
Placental volume (mm^3^)	214.80 ± 94.67
VI	14.38 ± 5.67
FI	43.27 ± 8.76
VFI	8.46 ± 4.20

NT: Nuchal Translucency; CRL: Crown-Rump Length; VI: Vascularization Index; FI: Flow Index; VFI: Vascularization Flow Index.

**Table 3 bioengineering-11-00509-t003:** Levels of HLA-G in samples of amniotic fluid and serum (N = 41) *.

sHLA-G concentration in amniotic fluid (ng/mL)	53.39 ± 19.00 ng/mL
sHLA-G concentration in serum (ng/mL)	51.05 ± 26.99 ng/mL

sHLA-G: soluble Human Leukocyte Antigen-G. * Continuous variables are displayed as mean ± standard deviation (SD).

**Table 4 bioengineering-11-00509-t004:** Correlation between maternal and sonographic data and levels of sHLA-G in maternal serum and amniotic fluid (N = 41).

	sHLA-G Level in Serum				sHLA-G in Amniotic Fluid			
	Univariate Linear Regression		Multivariate Linear Regression		Univariate Linear Regression		Multivariate Linear Regression	
	β	CI	β	CI	β	CI	β	CI
Clinical and obstetric characteristics								
Maternal age	0.01	−1.31–1.38	0.01	−1.71–1.74	−0.20	−1.51–0.39	−0.17	−1.75–0.79
Previous parity	−0.12	−13.18–6.00	−0.15	−16.17–7.50	−0.08	−8.54–5.33	0.08	−7.21–10.43
BMI at the time of genetic counseling (kg/m^2^)	0.19	−0.57–2.21	0.15	−0.82–2.16	−0.04	−1.22–0.93	0.00	−1.10–1.11
Birth weight (grams)	−0.02	−0.03–0.02	−0.01	−0.03–0.02	0.01	−0.02–0.02	−0.01	−0.02–0.02
Birth weight (percentile)	0.05	−0.30–0.40	0.18	−0.31–0.42	−0.02	−0.27–0.24	−0.00	−0.27–0.27
NT	−0.11	−17.87–8.79	−0.17	−22.29–8.50	−0.30	−17.78–0.86	−0.38 *	−21.73–0.04 *
CRL at NT	0.30	−0.08−2.52	0.76	−0.38−2.69	0.10	−0.71−1.32	0.14	−0.75−1.60
GA at the time of delivery	−0.11	−8.85−4.34	−0.06	−1.14−0.81	0.02	−4.45−5.08	0.03	−0.63−0.77
GA at the time of amniocentesis (weeks)	−0.13	−8.22−3.45	0.46	−1.30−0.55	0.24	−1.12−7.10	0.20	−0.31−1.02
Fetal sonography at the time of amniocentesis								
Head circumference (mm)	−0.12	−0.75−0.35	−0.09	−1.12−0.83	0.26	−0.08−0.69	0.19	−0.49−0.95
Head circumference (percentile)	−0.03	−0.32–0.28	−0.16	−0.51–0.21	−0.08	−0.27–0.17	0.08	−0.24–0.34
Abdominal circumference (mm)	0.01	−0.71–0.71	0.70	−0.30–2.77	0.33	−0.08–0.86	0.93	−0.14–2.32
Abdominal circumference (percentile)	0.41 *	−0.08–0.75 *	0.35	−0.07–0.84	0.26	−0.12–0.51	0.35	−0.08–0.61
Femur length (mm)	0.07	−2.54–1.74	−0.18	−7.88–5.78	0.23	−0.61–2.53	−0.28	−6.89–4.61
Femur length (percentile)	0.20	−0.18–0.59	−0.02	−0.49–0.45	−0.00	−0.30–0.29	−0.12	−0.49–0.31
Estimated fetal weight (grams)	−0.04	−0.16–0.14	0.64	−0.22–0.69	0.26	−0.04–0.17	0.73	−0.17–0.53
Estimated fetal weight (percentile)	0.41 *	−0.02–0.84 *	0.31	−0.15–0.85	0.24	−0.14–0.52	−0.31	−0.12–0.63
Placental sonography at the time of amniocentesis								
Placental volume (mm^3^)	0.02	−0.09–0.10	−0.03	−0.11–0.10	−0.09	−0.09–0.05	0.01	−0.07–0.08
VI	−0.10	−2.00–1.08	−0.16	−2.56–1.03	−0.34 *	−2.13–0.06 *	−0.38 *	−2.47–0.03 *
FI	0.05	−0.86–1.14	0.04	−0.92–1.17	−0.18	−1.12–0.34	−0.13	−1.06–0.50
VFI	−0.34 *	−3.58–0.46 *	−0.32	−4.32–0.20	−0.44 *	−3.28–0.63 *	−0.52 *	−3.79–0.72 *

sHLA-G: soluble Human Leukocyte Antigen-G; BMI: body mass index; GA: Gestational Age; VI: Vascularization Index; FI: Flow Index; VFI: Vascularization Flow Index. * *p* < 0.05.

## Data Availability

Data can be made available by the corresponding authors upon request.

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
