# Peer review of "How the Soluble Human Leukocyte Antigen-G levels in Amniotic Fluid and Maternal Serum Correlate with the Feto-Placental Growth in Uncomplicated Pregnancies"

_bioengineering, 2024, doi:10.3390/bioengineering11050509_

Round 1

Reviewer 1 Report

Comments and Suggestions for Authors

Comments to Authors

The authors investigated the relationship between sHLA-G in maternal serum, amniotic fluid with fetal and placental growth.

Here’s some comments for improvement:

Major

Overall

-                           There were significant similarities detected in a previously published paper by the same group of authors [1]. Both the papers studied on sHLA-G in maternal serum and amniotic fluid and their association with pregnancy outcomes. Besides the different in the timing of patients’ recruitment, what are the significant different between the two studies?

References [1]: Vincze M, Sikovanyecz J Jr, Molnár A, Földesi I, Surányi A, Várbíró S, Németh G, Sikovanyecz J, Kozinszky Z. Predictive Capabilities of Human Leukocyte Antigen-G and Galectin-13 Levels in the Amniotic Fluid and Maternal Blood for the Pregnancy Outcome. Medicina (Kaunas). 2024 Jan 1;60(1):85. 

Introduction

-                           HLA-G expression in cancer is not related to the title or aim of the study. Suggest to be more focus on the subject of interest and to remove the irrelevant paragraphs.

Materials and Methods

-                           I am not very sure about the definition of “healthy” pregnancy in this study, as all pregnancies that were recruited were at higher risk of getting fetal or chromosomal abnormalities compared to the general population.

-                           All 41 women had had amniocentesis performed. What were the results? Any chromosomal abnormalities detected? This needs to be clearly stated in the result section.

-                           What about those presented with preterm delivery? Or those with clinical or histological chorioamnionitis? Are those women included or excluded in the study?

Results

-                           There were as high as 37 out of 41 women were in the advanced maternal age group, by definition more than 37 years of age. I was surprised to learn that the mean maternal age was only 33.63 years. What was the age range in these women? Suggest to relook again at the data.

-                           What should be the normal range for HLA-G levels in maternal serum & amniotic fluid in normal healthy pregnancy that was published previously? Is the level the remain constant throughout the entire gestational age?

-                           Table 2 & 3 – suggest to include also expected/normal value for all relevant parameters for that gestational age.

Discussion

-                           4th paragraph –maternal serum HLA-G level was correlated significantly with estimated fetal weight and abd circumference in term of percentiles, but not in term of actual weight and measurement in grams and mm respectively. I am slightly skeptical to draw a conclusion suggesting that sHLA-G could serve as potential predictor for fetal growth anomaly from this result alone.

-                           2nd last paragraph - the entire study was based on only 41 samples. I do not agree that it should be regarded as “large number of samples” as portrayed by the authors. Suggest to modify the statement.

Minor

Introduction, 2nd paragraph – “ET” cells in the placenta. Do you mean EVT cells?

There are significant spelling and grammatical errors seen throughout the manuscript. E.g. “macrophages” was spelled “machrophages”, “trophoblasts” was spelled “throphoblasts”, “effected” or was it “effects”? and etc. Suggest a careful relook at the manuscript and perhaps an English editing service by native speaker.

Most of the cited references are more than 5 years ago. Suggest to cite latest (recent 5 years) publications unless those cited ones are landmark papers.

Comments on the Quality of English Language

There are significant spelling and grammatical errors seen throughout the manuscript. E.g. “macrophages” was spelled “machrophages”, “trophoblasts” was spelled “throphoblasts”, “effected” or was it “effects”? and etc. Suggest a careful relook at the manuscript and perhaps an English editing service by native speaker.

Author Response

We are very thankful for your comments. Please, find below our corrections.

The authors investigated the relationship between sHLA-G in maternal serum, amniotic fluid with fetal and placental growth.

Here’s some comments for improvement:

Major

Overall

-                           There were significant similarities detected in a previously published paper by the same group of authors [1]. Both the papers studied on sHLA-G in maternal serum and amniotic fluid and their association with pregnancy outcomes. Besides the different in the timing of patients’ recruitment, what are the significant different between the two studies?

References [1]: Vincze M, Sikovanyecz J Jr, Molnár A, Földesi I, Surányi A, Várbíró S, Németh G, Sikovanyecz J, Kozinszky Z. Predictive Capabilities of Human Leukocyte Antigen-G and Galectin-13 Levels in the Amniotic Fluid and Maternal Blood for the Pregnancy Outcome. Medicina (Kaunas). 2024 Jan 1;60(1):85. 

We agree with the referee that there are similarities between the two MS, but the studied sample, aims and the results provided in the present MS is different from the other article that has been already published. We are endeavoured to set up a study for checking the HLA-G levels in a highly selected population where all of the pregnant women with complications were left out (see detailed in the Materials and Methods), whereas all recruited pregnant women were presented in our other publication. One can speculate, that the best controls from perinatological point of view would be the pregnant women with age between 20 and 29 years, but the amniocentesis is so uncommon. In our former article a significant correlation was found between circulating sHLA-G level and placental volume and between amniotic sHLA-G level and placental perfusion in the whole population who underwent amniocentesis, while in our present MS the results are very different.

Introduction

-                           HLA-G expression in cancer is not related to the title or aim of the study. Suggest to be more focus on the subject of interest and to remove the irrelevant paragraphs.

We fully agree with you and the sentences relating to cancer have been excluded and our Introduction section is more focused now.

Materials and Methods

-                           I am not very sure about the definition of “healthy” pregnancy in this study, as all pregnancies that were recruited were at higher risk of getting fetal or chromosomal abnormalities compared to the general population.

Thank you for your comment. The expression ’healthy’ has been reworded to ‘uncomplicated’.

-                           All 41 women had had amniocentesis performed. What were the results? Any chromosomal abnormalities detected? This needs to be clearly stated in the result section.

The exclusion criteria are listed in the ’Materials and Methods’, and all of the chromosomal anomalies were excluded from the study. We think that the    exclusion and inclusion criteria are stated in the Materials and Methods section completely and we don’t think that we need to repeat this in the Result section.

-                           What about those presented with preterm delivery? Or those with clinical or histological chorioamnionitis? Are those women included or excluded in the study?

Your notification is important. Of course, these parturients were also left out from the study, and now the description of the sample is complete.

Results

-                           There were as high as 37 out of 41 women were in the advanced maternal age group, by definition more than 37 years of age. I was surprised to learn that the mean maternal age was only 33.63 years. What was the age range in these women? Suggest to relook again at the data.

Thank you for pointing to this fault. Advanced age definition was/is not > 37 but > 35 years of age. It has been changed throughout in the text. The age range was 18 to 42. 11 women were below 20. There are some young pregnant women in our sample since our aim was to collect only went through amniocentesis but at the same time had a fully uncomplicated pregnancy. However, we would like to emphasize that this is a sample of pregnant women without any pregnancy complication, and we think we have some novelties in our MS.

-                           What should be the normal range for HLA-G levels in maternal serum & amniotic fluid in normal healthy pregnancy that was published previously? Is the level the remain constant throughout the entire gestational age?

There are three publications on sHLA-G levels in maternal serum in the second trimester, and they are present similar results. Both articles are discussed in our Discussion section (Steinborn et al., 2003/2007 and Beneventi et al., 2016). Furthermore, there are two publications (Kusanovic et al., 2009 and Hackmon et al., 2004) on the amniotic fluid sHLA-G levels in the second trimester, but their data related to uncomplicated and complicated pregnancies as well. We think that our results on amniotic sHLA-G levels are interpreting the healthiest sample of pregnant women without any complications. The title of the present MS has been changed a little.

-                           Table 2 & 3 – suggest to include also expected/normal value for all relevant parameters for that gestational age.

Thank you for this comment. We don’t think that it is necessary to interpret the normal or average ranges of the values. We think that the median values are impractical to provide, because the pregnant women in our sample were between 16th and 22nd gestational weeks. The normal range of the percentiles is 10-90% and the range of percentiles reflect the distribution of our samples in terms of the variables.

Discussion

-                           4th paragraph –maternal serum HLA-G level was correlated significantly with estimated fetal weight and abd circumference in term of percentiles, but not in term of actual weight and measurement in grams and mm respectively. I am slightly skeptical to draw a conclusion suggesting that sHLA-G could serve as potential predictor for fetal growth anomaly from this result alone.

Thank you for your comment, but we did not state this in our MS. We did not include any pregnancies with large for gestational age or small for gestational age. Hence, our sample is not able to reveal how either the maternal circulating sHLA-G or the amniotic fluid sHLA-G level can predict the fetal growth anomaly. We have changed it in the text.

-                           2nd last paragraph - the entire study was based on only 41 samples. I do not agree that it should be regarded as “large number of samples” as portrayed by the authors. Suggest to modify the statement.

 We are very thankful for your comments. We excluded this sentence from our MS.

Minor

Introduction, 2nd paragraph – “ET” cells in the placenta. Do you mean EVT cells?

Yes, it has been changed.

There are significant spelling and grammatical errors seen throughout the manuscript. E.g. “macrophages” was spelled “machrophages”, “trophoblasts” was spelled “throphoblasts”, “effected” or was it “effects”? and etc. Suggest a careful relook at the manuscript and perhaps an English editing service by native speaker.

It was done.

Most of the cited references are more than 5 years ago. Suggest to cite latest (recent 5 years) publications unless those cited ones are landmark papers.

It has been changed as you proposed.

Comments on the Quality of English Language

There are significant spelling and grammatical errors seen throughout the manuscript. E.g. “macrophages” was spelled “machrophages”, “trophoblasts” was spelled “throphoblasts”, “effected” or was it “effects”? and etc. Suggest a careful relook at the manuscript and perhaps an English editing service by native speaker.

Our manuscript is reviewed.

Reviewer 2 Report

Comments and Suggestions for Authors  

The Authors discussed the prognostic role of soluble Human Leukocyte Antigen-G (sHLA-G) levels in amniotic fluid and maternal serum. They found a correlation between the levels of sHLA-G in amniotic fluid, placenta, and fetal biometry in the 2nd-3rd trimester and maternal serum levels, speculating that sHLA-G can be a helpful indicator concerning the well-being of pregnancy and fetus.

The paper is well-organized and written.

MINOR CONCERNS

In the Results section, there is a sentence that is not clear: "As expected, the birth weight of the offsprings very without extremities." The Authors should rephrase or expand the concept, as it is not understandable.

The tables' format is poorly readable (formatted as "in the center"). Please adjust the table content, possibly avoiding splitting a single word into two lines. 

Comments on the Quality of English Language

The quality of the English Language is overall very good.

Please check for minor inconsistencies.

Author Response

Thank you for your careful review. You will find our answers below.

The Authors discussed the prognostic role of soluble Human Leukocyte Antigen-G (sHLA-G) levels in amniotic fluid and maternal serum. They found a correlation between the levels of sHLA-G in amniotic fluid, placenta, and fetal biometry in the 2nd-3rd trimester and maternal serum levels, speculating that sHLA-G can be a helpful indicator concerning the well-being of pregnancy and fetus.

The paper is well-organized and written.

            We are grateful to your laudative comments.

MINOR CONCERNS

In the Results section, there is a sentence that is not clear: "As expected, the birth weight of the offsprings very without extremities." The Authors should rephrase or expand the concept, as it is not understandable.

We have rephrased our sentence. We would like to express that we recruited pregnant women only with normal birth weight and not with small for gestational age and large for gestational age.

The tables' format is poorly readable (formatted as "in the center"). Please adjust the table content, possibly avoiding splitting a single word into two lines. 

We have done an adjustment that makes that every line is more readable.

Comments on the Quality of English Language

The quality of the English Language is overall very good.

Please check for minor inconsistencies.

It was done.

Reviewer 3 Report

Comments and Suggestions for Authors

1. The Introduction is too long and very hard for the reader to follow; keep the introduction to the area of the study as cancer etc confuses the reader.

2. The results of the amniocentesis should be included for the 41 cases because if all were normal your screening test is poor

3. Hard to support your 'unraveled' NT comments with just 2 NT cases. 

4. You need some abnormal cases to support your growth correlations but seems to a good start. Table 2 needs a control value from local or national registries for comparative 'normality'.

5. Birth outcomes are required for the cohort as Table 4 just provides the time at assessment.

Comments on the Quality of English Language

Adequate.

Author Response

Thank you for your valuable comments.

Comments and Suggestions for Authors

  1. The Introduction is too long and very hard for the reader to follow; keep the introduction to the area of the study as cancer etc confuses the reader.

The Introduction has become significantly shorter, and the MS has been briefed.

  1. The results of the amniocentesis should be included for the 41 cases because if all were normal your screening test is poor.

This is a highly selected population, and the selection was based not only on genetic or structural malformations but on pregnancy complications also. All complicated pregnancies were excluded from the study, that is why the relatively young mothers (20-29 years) are potentially overrepresented and mothers at higher age (>35 years) are underrepresented in our sample. All the amniocentesis results were normal (euploid, genetically normal, with no minor or major malformation) as it is stated in the Materials and Methods. Thus, our sample is not reflecting at all our screening test capability since this sample is a highly selected. We have made some corrections in the description of our sample in order to be more understandable.

  1. Hard to support your 'unraveled' NT comments with just 2 NT cases.

Thank you. We are not drawing a conclusion of 2 cases, but we are drawing a conclusion on 41 cases. We are not talking about that the HLA-G level is good for NT based screening method for chromosomal abnormality, but we say instead that HLA-G is following the NT size expressing a negative correlation in healthy fetuses. This is an embryological/fetal anatomic question. However, these results require more studies to be supported. We have changed in the text.

  1. You need some abnormal cases to support your growth correlations but seems to a good start.

Thank you for the hint, our next step will be the analysis of the abnormal cases as well, but we have some significant results in a completely normal amniocentesis cases that we want to publish.

Table 2 needs a control value from local or national registries for comparative 'normality'.

It is practically impossible to provide or to interpret the median values because it is a range of a gestational length, and the data is not referring a time-point during the pregnancy. Furthermore, we think that it would be very hard to follow such a table, and the percentile range of some variables reflect somehow the range of the distribution of most of the

  1. Birth outcomes are required for the cohort as Table 4 just provides the time at assessment.

Thank you for the comment, but we rather think that it is not needed since we have a high number of variables compared to the size of our sample. If you or the Editor think it is needed, then we provide these values. In addition, the overwhelming subsample delivered vaginally with no antepartum or postpartum complication.

Comments on the Quality of English Language

Adequate

Reviewer 4 Report

Comments and Suggestions for Authors

Angiogenic factors derived from the trophoblast are recognized as pivotal players in the pathogenesis of various pregnancy complications. Human Leukocyte Antigen-G (HLA-G), a member of the non-classical human major histocompatibility complex (MHC-I) molecules, exists in both membrane-bound and soluble forms. It is predominantly expressed by extravillous cytotrophoblasts situated at the placental interface between maternal and fetal compartments, crucially contributing to immune tolerance.

This study aimed to investigate the association between concentrations of soluble Human Leukocyte Antigen-G (sHLA-G) in maternal serum and amniotic fluid at 16-22 weeks of gestation and sonographic measurements of fetal and placental growth. In a sample of 41 uncomplicated singleton pregnancies, sHLA-G levels in serum and amniotic fluid were assessed alongside fetal biometric data and placental volume and perfusion indices. A sandwich enzyme-linked immunosorbent assay (ELISA) kit was employed to quantify sHLA-G levels (U/ml). Notably, while the study sample size may be considered low given the number of parameters examined, it is worth noting the lack of a sample size calculation to justify this number.

The authors found that sHLA-G levels remained stable in both amniotic fluid and serum throughout mid-pregnancy. Positive correlations were observed between serum sHLA-G levels and amniotic sHLA-G levels. Furthermore, significant associations were found between serum sHLA-G levels and various fetal biometric measurements. Conversely, amniotic sHLA-G levels exhibited negative correlations with placental perfusion. Notably, a decreased level of amniotic sHLA-G was significantly linked to nuchal translucency.

The Authors conclude that assessment of sHLA-G in amniotic fluid may serve as a potential indicator of placental function, while serum sHLA-G levels could offer prognostic insights into feto-placental insufficiency. However, further studies with larger sample sizes and proper sample size calculations are warranted to validate these findings.

Author Response

Thank you for your laudative comments.

Angiogenic factors derived from the trophoblast are recognized as pivotal players in the pathogenesis of various pregnancy complications. Human Leukocyte Antigen-G (HLA-G), a member of the non-classical human major histocompatibility complex (MHC-I) molecules, exists in both membrane-bound and soluble forms. It is predominantly expressed by extravillous cytotrophoblasts situated at the placental interface between maternal and fetal compartments, crucially contributing to immune tolerance.

This study aimed to investigate the association between concentrations of soluble Human Leukocyte Antigen-G (sHLA-G) in maternal serum and amniotic fluid at 16-22 weeks of gestation and sonographic measurements of fetal and placental growth. In a sample of 41 uncomplicated singleton pregnancies, sHLA-G levels in serum and amniotic fluid were assessed alongside fetal biometric data and placental volume and perfusion indices. A sandwich enzyme-linked immunosorbent assay (ELISA) kit was employed to quantify sHLA-G levels (U/ml). Notably, while the study sample size may be considered low given the number of parameters examined, it is worth noting the lack of a sample size calculation to justify this number.

We have quite a lot of parameters, but it was difficult to find adequate study persons in our centre.

The authors found that sHLA-G levels remained stable in both amniotic fluid and serum throughout mid-pregnancy. Positive correlations were observed between serum sHLA-G levels and amniotic sHLA-G levels. Furthermore, significant associations were found between serum sHLA-G levels and various fetal biometric measurements. Conversely, amniotic sHLA-G levels exhibited negative correlations with placental perfusion. Notably, a decreased level of amniotic sHLA-G was significantly linked to nuchal translucency.

The Authors conclude that assessment of sHLA-G in amniotic fluid may serve as a potential indicator of placental function, while serum sHLA-G levels could offer prognostic insights into feto-placental insufficiency. However, further studies with larger sample sizes and proper sample size calculations are warranted to validate these findings.

Many thanks for your concerns.

Round 2

Reviewer 2 Report

Comments and Suggestions for Authors

The Author correctly addressed the Reviewer's suggestions.

However, there is a too large overlap with the Authors' precedent paper.

Even if I understand the reason why, I would suggest a careful revision of the possible overlap, and I strongly recommend resolving them.

Author Response

We performed a careful check of our database, but the population is different with different recruited names of the subjects.

Reviewer 3 Report

Comments and Suggestions for Authors

Thankyou for the update and will accept your answers to my concerns and it may be that the cohort size limits some clinical clearity.

Author Response

Thank you for your nice comments.

Reviewer 4 Report

Comments and Suggestions for Authors

No further comments

Author Response

Thank you!

Round 3

Reviewer 2 Report

Comments and Suggestions for Authors

I understand that although the population is different. I referred to the fact that a large part of the material and methods are identical to one of your prior articles.